# POINT-AND-CLICK: A PROCEDURAL BENCHMARK FOR 2D ADVENTURE PUZZLE SOLVING

## ABSTRACT

Point-and-click adventure games offer an ideal platform for testing multimodal large language model agents on long-horizon reasoning, commonsense knowledge, and language-perception grounding. Such games demand creative, compositional reasoning and the deduction of implicit goals. However, existing benchmarks provide limited support for compositional and generative puzzles, and often suffer from data contamination. To bridge this gap, we present Point-and-Click, a benchmark for 2D adventure games that procedurally generates rich puzzles and provides ground-truth solutions for evaluation. The environment instantiates controllable directed acyclic graphs of puzzle dependencies over primitives like keys/locks, codes, and pattern matching, spanning an exponentially scaling number of layouts with tunable difficulty. Experiments reveal the limitations of current multimodal LLM/VLM agents on this benchmark. We hope Point-and-Click serves as a rigorous testbed for progress on general-purpose embodied reasoning and implicit goal deduction in interactive environments.

## 1 INTRODUCTION

Humans excel at solving complex puzzles in interactive environments by combining **long-horizon planning**, **commonsense knowledge**, and **perception-grounded reasoning**. A classic example is the point-and-click adventure game, where a player must explore a scene, collect and combine objects, and deduce how to use them to achieve an implicit goal (e.g. escaping a room). Such games require the player to interpret visual cues, recall or acquire knowledge about object uses, and plan multi-step solutions – all without an explicit instruction. They therefore present an ideal challenge for multimodal intelligent agents that aim to mimic human problem-solving.

Recent advances in large language models (LLMs) and vision-language models (VLMs) have yielded agents with impressive capabilities in language and vision understanding. However, it remains unclear whether these models possess the general reasoning ability to solve interactive puzzles that require chaining many steps and inferring hidden objectives. Existing benchmarks only scratch the surface of this question. Many focus on single-step question answering or short-horizon tasks, rather than the creative, compositional reasoning required in puzzles (Chia et al., 2024; Wang et al., 2025b). Other benchmarks, while focusing on long-chain puzzle solving, are often built from existing games or static (Ahn et al., 2025; Lim et al., 2025). This risks data contamination as LLMs pretrained on large-scale internet content may have memorized solutions to published puzzles, or fails to provide controllable diversity and sufficient scaling necessary to evaluate generalization. This paper addresses these gaps by introducing Point-and-Click, a new generative benchmark designed to rigorously evaluate multimodal agents on complex puzzle-solving. The overview of this benchmark is shown in Figure 1.

In Point-and-Click, each task is a **procedurally generated** 2D adventure game room containing a network of interdependent puzzles. The agent (or player) must discover and solve these puzzles to ultimately achieve an implicit goal (such as unlocking the exit door). The environment is built to test several key abilities:

- Long-horizon planning: Puzzles are compositional – multiple items and clues must be found and combined in sequence to reach a solution, often spanning 10+ to 100+ steps.

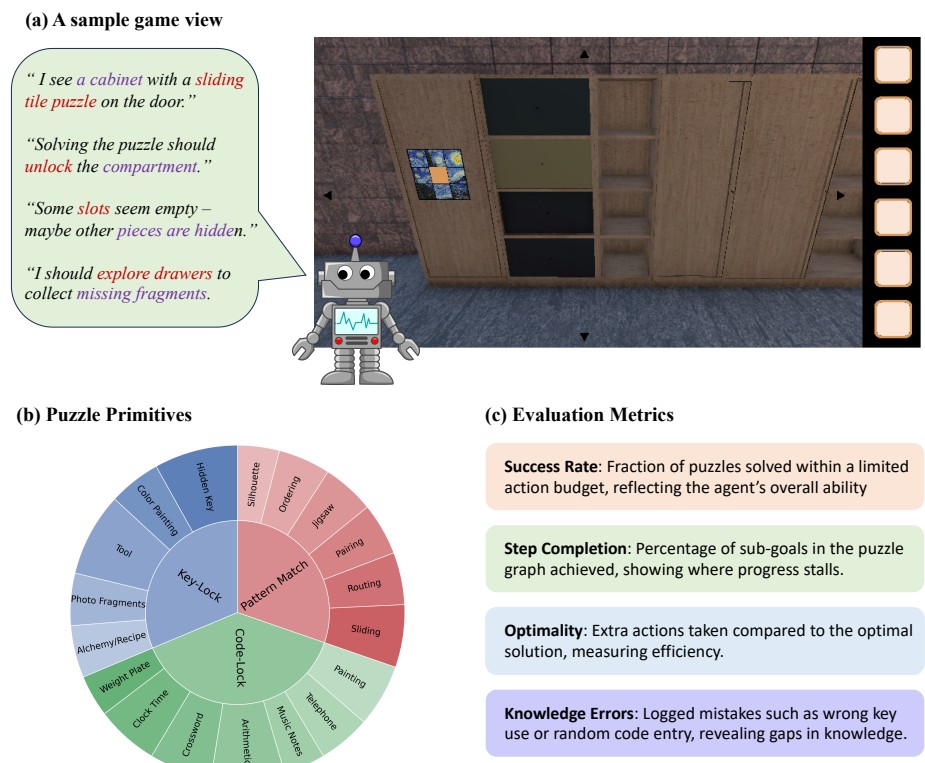

Figure 1: Overview of the Point-and-Click benchmark.

- Commonsense and factual knowledge: The agent must leverage basic knowledge about everyday objects and their affordances (e.g. keys open locks, combinations of numbers unlock codes) to decide plausible actions.

- Language–perception grounding: Clues may be visual (pattern on a painting) or textual (a written note with a code). The agent needs to interpret visual signals and map them to game actions, grounding linguistic reasoning in perception.

Crucially, the goals in Point-and-Click are **implicit**. Unlike instruction-following tasks (e.g. AL-FRED (Shridhar et al., 2020) where a directive is given), the agent is not told what to achieve. It must infer the objective (usually to access a locked reward or escape) by exploring the environment and recognizing what final state would constitute "success." This implicit goal deduction is a hallmark of human puzzle-solving and a challenging new test for AI.

Our benchmark generates puzzles procedurally to ensure endless variety and zero contamination. Each puzzle instance is defined by a ground-truth causal graph (a DAG) connecting intermediate subgoals. For example, a causal graph may specify that unlocking the door requires finding a key, which in turn require other steps (decipher a clue, enter a code, etc.). By varying the graph structure and instantiating it with different objects/locations, we obtain a theoretically infinite set of puzzles that an agent cannot memorize in advance. The provided ground-truth graphs allow fine-grained evaluation of an agent's performance (e.g. did it solve specific sub-puzzles, in what order, with which errors) rather than a coarse success/failure only.

In the experiments, we observe a sharp human–agent gap: the best model reaches 40/10/0% success versus humans at 100/96/64% under our simple/medium/hard experiment settings. Agents often make partial progress without completion, and CUA is the most action-efficient when progressing Failures stem mainly from perception/attention misses, brittle riddle solving, and forgetting clues. These results reveal a pronounced difficulty cliff and motivate tighter perception–reasoning integration, persistent memory, and explicit planning over DAG-structured sub-goals.

**Contributions.** We introduce Point-and-Click, a procedural benchmark for evaluating multimodal agents on long-horizon puzzle-solving in 2D adventure games. (1) Unlike prior benchmarks, Point-and-Click generates a theoretically unbounded set of interactive puzzle rooms defined by controllable dependency graphs over core primitives such as keys/locks, numeric codes, and visual patterns. (2) Each instance comes with a ground-truth causal graph, enabling fine-grained evaluation beyond binary success metrics. (3) Our benchmark emphasizes implicit goal inference, compositional reasoning, and language-perception grounding, posing a rigorous challenge for current LLM- and VLM-based agents. (4) Experiments show that state-of-the-art models struggle to solve even moderately complex puzzles, highlighting the benchmark's difficulty and its potential as a testbed for research in embodied reasoning, planning, and commonsense understanding.

## 2 RELATED WORK

**Benchmarks for Complex Reasoning.** A variety of benchmarks have been proposed to evaluate advanced reasoning in both language and multimodal settings. Some focuses on abstract visual puzzles with varying patterns based on colors, numbers, shapes, sizes, etc. (Chia et al., 2024; Estermann et al., 2024; Ghosal et al., 2025; Chollet et al., 2025). These works are further complemented with large-scale puzzlehunt benchmarks such as EnigmaEval (Wang et al., 2025a) and PuzzleWorld (Li et al., 2025) which curate complex puzzles from real competitions. Compared to traditional benchmarks, puzzle-based evaluations probe multi-step deductive reasoning and the synthesis of multimodal clues. Even state-of-the-art model achieves only $\leq 7\%$ accuracy on EnigmaEval's normal split, despite saturating easier tasks, underscoring the need for benchmarks that test long-horizon vision–language reasoning beyond static QA or short-context settings.

**Interactive Fiction and Escape-Game Environments.** Research in interactive fiction (IF) games has long informed the design of complex puzzle environments. Classic text adventures pioneered open-ended puzzle solving via natural language. Jericho (Hausknecht et al., 2020) provides dozens of human-written IF games (such as Zork) and challenges agents with combinatorial action spaces and commonsense reasoning. Similarly, Microsoft's TextWorld (Côté et al., 2018) enables generation of text-based games with controllable difficulty and state tracking, allowing systematic evaluation of an agent's ability to solve adventure games through textual commands. This line of research sparked a surge of subsequent work (Urbanek et al., 2019; Tan et al., 2023; Ma et al., 2024; Qian et al., 2025; Phan et al., 2025). Building on text-only adventures, recent efforts integrate visual and embodied elements to create escape-room style challenges. Obstacle Tower (Juliani et al., 2019) presents a procedurally generated 3D environment where agents learn from pixels under sparse rewards to traverse a multi-level tower. ALFWorld (Shridhar et al., 2021) aligns TextWorld puzzles with the embodied ALFRED tasks, enabling agents to transfer abstract language policies to visual tasks. Recent advancements such as EscapeCraft (Wang et al., 2025b), VisEscape (Lim et al., 2025) and FlashAdventure (Ahn et al., 2025) provide 2D/3D room-escape environment where agents must explore virtual rooms, recognize objects, and use tools to unlock exits. However, these static benchmarks are built from published games, suffering from contamination and memorization as models can recall solutions from pretraining data, or lack controllable diversity or scaling, limiting their ability to test generalization and reasoning.

**Procedural Puzzle and Content Generation.** Procedural content generation (PCG) has been widely explored for both puzzles and environments, with growing focus on narrative-driven adventures. Early systems such as Puzzle-Dice (Fernández-Vara & Thomson, 2012) model puzzles as dependency graphs of design patterns, enabling replayable point-and-click games, while planning-based approaches such as Dart & Nelson (2012) model items as "smart terrain" with causal effects and insert these items into existing game environments. More recent work such as SPHINX (Morgan & Haahr, 2020) uses grammar-based rules to scale puzzle generation with greater expressiveness and content variety. Complementary research has tackled environment generation, from graph-grammar–based dungeon generation (Dormans, 2010; De Kegel & Haahr, 2019) to large-scale photorealistic 3D room scene synthesis (Raistrick et al., 2024; Zhou et al., 2025). Across these efforts, evaluation emphasizes solvability, variety, and user engagement. Together, this literature highlights how procedural puzzle and content generation can provide scalable, diverse, and rigorous testbeds for evaluating reasoning in adventure-style games, directly motivating our benchmark.

## 3 THE POINT-AND-CLICK BENCHMARK ENVIRONMENT

In this section, we elaborate on the details of the Point-and-Click environment. We introduce the problem formulation, basic components of the environment, and the procedural generation mechanism to create puzzles. The design principle is to synthesize compositional puzzles that require commonsense knowledge and multiple steps to solve, parameterize difficulty to test agents of varying skill, and avoid fixed data that an agent could exploit via prior pretrain knowledge.

### 3.1 PROBLEM FORMULATION

We model Point-and-Click as a partially observable Markov decision process (POMDP) $M =< S, A, O, \Omega, T, R >$. The hidden state $s \in S$ encodes the puzzle's dependency DAG $G = (V, E)$ with node statuses, object/container flags and relations, code/pattern parameters, and agent inventory. Actions are pure GUI mouse inputs: $A = (x, y)$ where $(x, y)$ are screen coordinates in normalized image space. The engine maps clicks to affordance-triggered interactions (e.g., opening a container, picking up an item, using an item on a target). After $a_t \in A$, the environment transitions according to $T(s_{t+1}|s_t, a_t)$, updating object states and advancing subgoals when preconditions are satisfied. The agent then receives an observation $o_{t+1} \in \Omega$ drawn from $O(o_{t+1}|s_{t+1}, a_t)$. In our benchmark $\Omega$ is the RGB framebuffer of the current view (including UI components such as navigate buttons and inventory pixels), so observations are partial until the agent reveals hidden content. Dynamics are deterministic by default. The reward is dense over subgoals: let $\Delta z_{t+1}$ be the set of DAG nodes whose status transitions to `solved` at $t + 1$, then

$$R(s_t, a_t) = \sum_{v \in \Delta z_{t+1}} r_v + \mathbf{1}[z_{v^\star} = \texttt{solved}]r_{\text{goal}},$$

with $r_v > 0$ per achieved subgoal and a larger terminal bonus $r_{\text{goal}} \gg r_v$ when the goal node $v^\star$ completes. Episodes end when a goal node $v^\star$ is solved or a budget $T_{\max}$ is reached. The agent maximizes $\mathbb{E}[\sum_{t=0}^{\infty} \gamma^t R(s_t, a_t)], \gamma \in [0, 1)$. This formulation covers VLM-based GUI agents, model-free RL from pixels, and hybrid planners that reason over beliefs about $G$ and object states.

### 3.2 ENVIRONMENT BASICS

Point-and-Click presents an interactive environment that emulates the perceptual and action modalities of human gameplay in point-and-click adventure games. At each step, the agent receives a 2D visual observation and produces a mouse-click action as output.

The visual input is a rendered 2D scene of a single room from fixed perspectives. Each room contains interactive objects such as items, clues, and containers. For example, a generated room might visually depict a kitchen with cabinets, a locked box on a table, a painting on the wall, etc., depending on the puzzle. Every object maintains an internal state (e.g., a box may be locked or unlocked, a painting might conceal a secret code, and an item may be intact or broken, etc.).

Actions are expressed entirely through mouse clicks, which can be functionally categorized as follows:

- **Examine [object]**: Inspect an object to obtain more details, such as zooming into the image of a painting and revealing a hidden clue. This action tests perception and possibly yields additional information.

- **Pick up [object] / Use [object] on [target]**: The agent can pick up portable items (adding to the inventory) and later use or combine them with other objects. For instance, first clicking the key in the inventory then clicking the box attempt to unlock the box, or combining liquid A with liquid B to obtain liquid C.

- **Open / Close / Move [object]**: Certain objects like doors, containers (drawers, cabinets) can be opened or moved if not locked.

- **Enter [code]**: Input numeric or symbolic codes into interfaces such as keypads or safes by clicking and changing the combination.

- **Other**: Perform context-specific interactions not captured above.

For each puzzle instance, the environment retains the ground-truth DAG of dependency relations and the mapping to actual in-game events. During evaluation, the agent's interactions are monitored against the causal graph. This enables fine-grained evaluation metrics beyond simply "solved or not." We define several metrics:

- **Success Rate**: The fraction of puzzles in a full benchmark suite where the agent successfully achieved the end goal within a given action budget . We prevent agents from brute-forcing indefinitely by assigning a time limit proportional to the ground-truth solution length. This is the primary measure of an agent's overall ability.

- **Step Completion**: We report the percentage of sub-goals in the causal graph achieved. This helps pinpoint where the agent failed, e.g., it managed to do early steps but got stuck on a particular type of puzzle.

- **Optimality**: We compare the agent's action sequence to the optimal reference solution and count how many extra actions were taken. This measures efficiency and whether the agent wasted time on irrelevant actions.

- **Knowledge Errors**: We log specific mistakes, which indicate either lack of knowledge or exploration strategy. For example, if an agent tries to use a wrong key on a lock repeatedly, or enters random codes, that would be recorded.

### 3.3 PROCEDURAL PUZZLE GENERATION

The core innovation of Point-and-Click is the puzzle generator, which creates a new puzzle instance by sampling a dependency DAG and instantiating it with concrete objects and clues. The workflow is illustrated in Figure 2. At a high level, the generator works as follows:

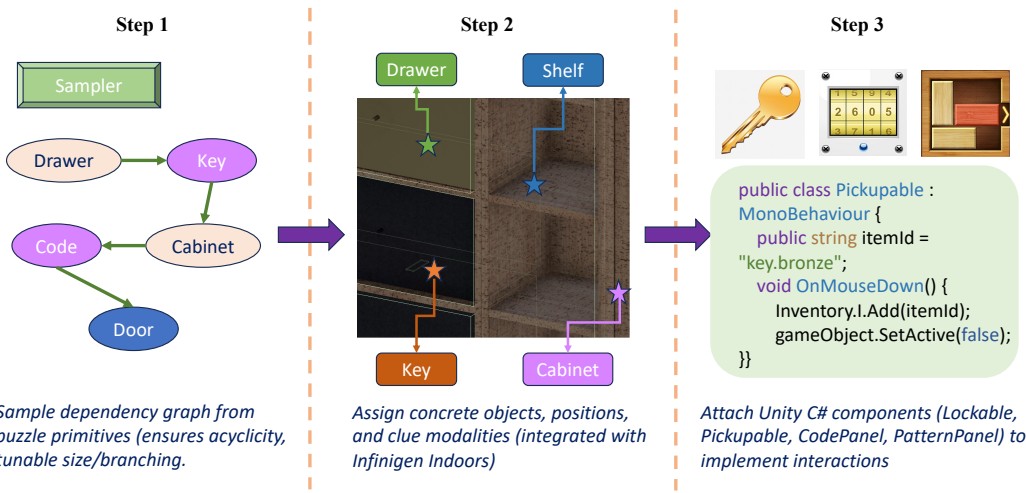

Figure 2: Puzzle generation workflow in Point-and-Click. (1) Sample a dependency DAG from puzzle primitives (Key–Lock, Code–Lock, Pattern Match); (2) instantiate objects and layout within a procedurally generated room; (3) attach Unity components to implement point-and-click interactions. This pipeline ensures coherence, solvability, and controllable difficulty.

1. **Sample a puzzle DAG**: We define a library of puzzle primitives, each representing a basic step or mechanism with specific requirements and outcomes. For example, a Key–Lock primitive requires a key item and a locked object; solving it unlocks the object and may yield a new item. A puzzle is structured as a DAG $G(V, E)$, where each node $v \in V$ represents a puzzle step and each edge $A \to B$ indicates that the outcome of $A$ enables $B$. For instance, a linear DAG might be: key opens box $\to$ box contains clue $\to$ clue is code to open door. More complex DAGs can include parallel sub-puzzles and merging branches. We ensure acyclicity for solvability, and control the DAG's size and structure

via difficulty parameters. Available primitives can be divided into the following categories, with illustrations in Figure 3:

- **Key–Lock**: A key item that unlocks a locked object (door, box, etc.). The "key" item is an abstract concept which means that opening the lock requires applying collectable items. The "key" could be a combination of multiple components, e.g., finding several fragments of a painting to form a full picture; mixing ingredients to make a tool.

- **Code–Lock**: A code (number/word/symbol) that opens a locked safe or door when entered. To open this puzzle, the player needs to observe visual clues to deduce the correct combination of code, but there is no need to collect any item into the inventory.

- **Pattern Match**: A visual or logical pattern that must be recognized (e.g. arranging symbols in the correct order, or matching a sequence). This puzzle is self-contained - the player should be able to solve it without knowledge of any other puzzles.

2. **Instantiate objects and layout**: Once a puzzle graph is sampled, the generator assigns concrete objects to each abstract node. For example, if one node is a Key–Lock puzzle, we might choose "key" and "locked cabinet" as the instantiation. If another node is a code puzzle, we might decide the code is a 4-digit number and hide it as a pattern in a painting on the wall. The generator has lists of possible items, locations (wall, floor, inside furniture), and hint modalities (text notes, visual patterns, riddles) to choose from. This generation process is built upon Infinigen Indoors (Raistrick et al., 2024) to leverage the automatic room layout procedure. Puzzle objects are implemented as custom assets with varying parameters that specify the details of the puzzle. If two puzzle nodes are connected in the DAG, their physical representations are linked accordingly using the constraint system. E.g., if unlocking box yields a clue for a code, the clue item (a note) is placed inside the box object in the environment.

3. **Implement interaction logic**: Each object instantiated in the environment is augmented with interactive behavior through Unity C# scripts. The behaviors are bound to a pre-defined set of reusable components that implement point-and-click affordances and state transitions. Concretely, we attach `Pickupable` (adds/removes an item from the inventory grid), `Lockable` (finite-state machine with `locked→unlocked`) to doors, drawers, and containers, `CodePanel` (token buffer + validator UI) to keypads/safes; and `PatternPanel` (grid/slider widgets with constraint checks) to pattern puzzles. The interaction system supports condition checking (e.g., does the player possess the required item?), state transitions (e.g., unlocking an object permanently), and feedback rendering (e.g. revealing a hidden clue). An inventory UI on the screen allows players to manage items, while in-world objects are annotated with metadata to control their behavior (e.g., clickable region, lock status, associated DAG node). Critically, all interactions are automatically derived from the underlying puzzle DAG, ensuring that the physical behavior of the environment matches the abstract logical structure, and that every game instance is fully solvable without manual scripting. This supports scalable generation of interactive puzzle rooms with guaranteed coherence and solvability.

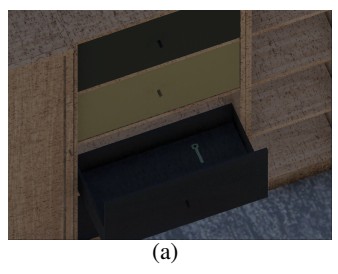
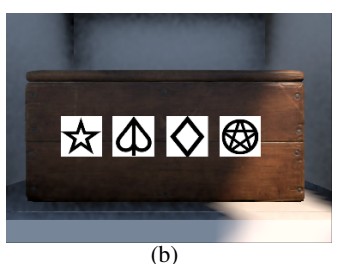
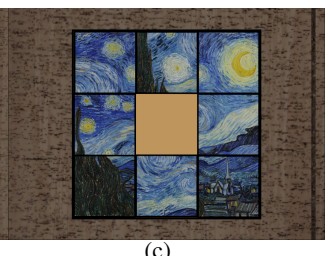

(a)          (b)          (c)

Figure 3: Examples of puzzle primitives used in Point-and-Click. (a) Key–Lock: a bronze key hidden in a drawer that unlocks a cabinet; (b) Code–Lock: a safe box requiring the correct symbol code deduced from visual clues; (c) Pattern Match: a sliding block puzzle where the player must complete a visual pattern. These primitives form the building blocks of sampled puzzle DAGs.

One of the key advantages of our approach is its **controllability**. The generator exposes parameters that allow fine-grained adjustments of puzzle complexity and structure. We can vary the number of steps by scaling the DAG size, tune the branching factor to produce linear or parallel puzzle chains, and include or exclude specific puzzle primitives for ablation studies. This flexibility enables systematic evaluation across different settings, as well as curriculum-style protocols where agents progress from simple to increasingly complex puzzles. Because puzzles are generated procedurally, agents can be tested on an effectively unlimited stream of novel episodes, mitigating memorization or overfitting. For standardized benchmarking, we also release a fixed evaluation suite of 30 puzzle rooms at varying difficulty levels, generated with held-out seeds and unpublished solutions to ensure fairness and reproducibility.

In summary, Point-and-Click offers a scalable and rigorous framework for studying embodied reasoning in interactive puzzle environments. By combining controllable generation, guaranteed solvability, and diverse puzzle primitives, it creates a challenging yet analyzable testbed. This allows researchers to probe fundamental capabilities of multimodal LLMs and RL agents, including long-horizon planning, commonsense reasoning, and implicit goal inference, while providing a standardized benchmark for fair comparison and reproducible progress.

## 4 EXPERIMENTAL RESULTS

### 4.1 EXPERIMENT SETUPS

We evaluate four agents in our Point-and-Click environment: (1) OpenAI's Computer Using Agent (OpenAI, 2025), representing a hybrid-reasoning model designed for onscreen control, (2) Claude-Sonnet-4.5 with Computer-Use enabled (Anthropic, 2024), another commercial model which performs goal-directed GUI actions, (3) UI-TARS-1.57B (Qin et al., 2025), an open-source vision-language UI agent, and (4) a human baseline. For model configuration, Claude-3.7-Sonnet runs with its computer-use/agentic interface; CUA uses the official tools-computer-use API; UI-TARS follows its public release settings; and for the human baseline, participants receive identical instructions and click budgets as agents.

Each agent is tested on a common set of 30 puzzle instances over three difficulty splits (simple, medium, hard, with 10 puzzles each of difficulty levels from 10 steps up to 100 steps) under a fixed action budget of 10 times of the ground-truth solution steps. We report three metrics per split: Success Rate (fraction of puzzles solved within budget), Step Completion (percentage of DAG subgoals achieved), and Optimality (extra actions over the reference solution). This setup aligns with prevailing UI-agent benchmarking practice emphasizing end-to-end task success under constrained interaction budgets.

### 4.2 BENCHMARKING RESULTS

Table 1 summarizes the performance metrics for each agent type. Here are the key observations:

- **Overall Success**: On *Simple* puzzles, OpenAI CUA leads with 40% success, outperforming Claude-3.7-Sonnet (20%) and UI-TARS-1.5-7B (10%). On *Medium*, both CUA and Claude tie at 10%, while UI-TARS drops to 0%. On *Hard*, all models achieve 0% success. Humans remain far ahead (100/96/64% across Simple/Medium/Hard).

- **Partial progress (Step).** CUA is strongest on *Simple* (53.00%), and retains the highest partial progress on *Hard* (12.30%), indicating it often advances several sub-goals even when it fails the full puzzle. Claude edges out others on *Medium* with the top Step score (25.80% vs. CUA 23.20%), suggesting mid-puzzle stalls rather than complete breakdowns. UI-TARS consistently trails (11.00/6.20/2.70%).

- **Efficiency (Optimality).** When models succeed or make progress, CUA is generally the most efficient: it has the best (lowest) extra-action counts on *Simple* (7.62) and *Medium* (9.95), narrowly beating Claude (9.96). On *Hard*, all methods hit the evaluation cap (10.00), consistent with timeouts or thrashing near dead-ends.

- **Difficulty cliff.** Moving from *Simple* to *Medium* produces a sharp drop: CUA falls from 40% to 10% success (–30 points), Claude from 20% to 10% (–10), and UI-TARS to 0%.

| Model | Difficulty | Success ↑ | Step ↑ | Opt. ↓ |
|---|---|---|---|---|
| Claude-3.7-Sonnet (Computer-Use) | Simple | 20% | 31.00% | 8.87 |
| | Medium | 10% | **25.80%** | 9.96 |
| | Hard | 0% | 5.20% | 10.00 |
| OpenAI CUA | Simple | **40%** | **53.00%** | **7.62** |
| | Medium | 10% | 23.20% | **9.95** |
| | Hard | 0% | **12.30%** | 10.00 |
| UI-TARS-1.5-7B | Simple | 10% | 11.00% | 9.87 |
| | Medium | 0% | 6.20% | 10.00 |
| | Hard | 0% | 2.70% | 10.00 |
| Human Performance | Simple | 100% | 100.00% | 3.13 |
| | Medium | 96% | 98.40% | 4.29 |
| | Hard | 64% | 70.60% | 5.58 |

Table 1: Point-and-Click benchmark results for agents across three difficulty tiers (Simple/Medium/Hard). Metrics: Success (fraction of puzzles solved, higher is better), Step (sub-goal completion rate, higher is better), and Optimality (extra actions vs. reference, lower is better).

Despite 0% success on *Hard*, non-zero Step scores (e.g., CUA 12.30%, Claude 5.20%) confirm that agents frequently make early progress but fail to complete multi-step dependencies.

- **Error Types**: We recorded that the LLM agents rarely made outright knowledge errors like using a wrong key on a lock more than once, indicating that they usually understood such concepts. Their errors were more from missing a hidden object or clue (perception/attention error) or from not reasoning through a riddle. Another common failure is forgetting a discovered clue after a few turns (context window issue), causing it to search again needlessly.

From these results, we can clearly see that Point-and-Click exposes a pronounced gap between current agents and humans. Even the strongest model (OpenAI CUA) solves only 40% of Simple puzzles and 10% of Medium, with 0% on Hard, while humans maintain high performance (100/96/64%). Step-level signals show that models often advance several sub-goals before stalling, and CUA is comparatively efficient when it progresses, yet all systems struggle to sustain long-horizon, dependency-laden plans under partial observability. Closing this gap will likely require stronger perception–reasoning integration (e.g., reliable counting and symbol grounding), persistent scratchpads or episodic memory to avoid revisiting solved clues, and explicit planning/search over DAG-structured sub-goals rather than myopic, step-by-step heuristics.

### 4.3 CASE STUDY

To illustrate typical behaviors behind the aggregate numbers in Table 1, we highlight three representative cases (full transcripts in the Appendix).

**Case 1: 12-Step Key Puzzle (Success by Claude-Sonnet-4.5).** The puzzle requires: find a key in on the wall, use the key to unlock a chest, inside chest find a new key, apply the key to unlock the door, with 12 clicks in total. Claude-Sonnet-4.5 handled this quite well. It navigated the room to find the key, picked it up, unlocked the chest, reasoned that the key was used to unlock the door. This shows that given a straightforward puzzle, Claude-Sonnet-4.5's general knowledge, reasoning, and grounding accuracy suffice. The Although successful, it was *not* strictly optimal: optimal actions were 12, while Claude-Sonnet-4.5 executed 67 (i.e., ×4.58 extra), reflecting the efficiency gap we see on average for Simple puzzles. This example typifies Claude-Sonnet-4.5's Simple-set profile: comparatively strong success among models yet still incurring noticeable extra actions.

**Case 2: 45-Step Puzzle (Failure of Claude-3.7-Sonnet at final step).** Here the agent had to: (1) gather two ingredients, (2) craft a tool, (3) reveal a hidden compartment, (4) read a hint to open a safe, (5) use the safe's key to exit. Claude-3.7-Sonnet (Computer-Use) progressed reliably

through early steps—collecting items, crafting, and opening the safe—reaching the exit with the key in inventory. However, it failed to perform the final key–door action before timeout, instead re-inspecting previously seen objects. Step completion was *93%*, consistent with our observation that Claude attains the highest *Medium* Step score (25.80% on average) despite a modest success rate (10%). The failure mode aligns with "late-stage stalls": partial plans are executed, but goal completion is missed without persistent goal tracking.

**Case 3: Hard puzzle with symbolic perception (OpenAI CUA misuses visual cue).** In a Hard configuration, a painting displayed four symbols indicating a directional combination for a lock. OpenAI CUA correctly perceived the symbols but treated "↑↓←→" as a literal string rather than a sequence of directional actions. After several incorrect entries and exploratory detours, it timed out having completed *27/102* sub-steps (26.5%). This mirrors the aggregate Hard-set pattern: *zero* end-to-end success across models, yet non-zero partial progress (CUA Hard Step $= 12.30\%$), indicating that agents can perceive salient cues but often fail to ground them into the correct action semantics over longer dependency chains.

## 5 DISCUSSION

**Implications for model design.** The failure modes suggest several concrete directions: (i) *Perception–reasoning integration*: agents need more reliable symbol grounding (e.g., mapping arrows or pictograms to action programs) and better object-centric perception to avoid missing small or occluded clues. (ii) *Persistent memory and state tracking*: maintaining a scratchpad or episodic memory over discovered clues and unresolved subgoals can reduce revisitation and forgetting. (iii) *Structured planning over DAGs*: explicit search or policy sketches that reason over hypothesized subgoal graphs (even when latent) may help bridge long horizons; lightweight belief updates over the latent $G$ can prioritize information-gathering and subgoal completion. (iv) *Goal management*: agents that continuously monitor goal completion conditions (Case 2) and maintain a to-do stack are less likely to miss terminal actions. (v) *Exploration with affordances*: learning affordance models ("what can be done where") can prune action spaces and guide clicks toward informative regions, improving both success and optimality.

**Leakage-Safe Structural Signals.** Although our evaluation hides ground-truth graphs from agents, the structure can shape training signals without trivializing the task. For instance, subgoal completions can provide auxiliary rewards for RL from pixels; graph-consistent trajectories can supervise imitation or behavior cloning; and latent-graph prediction losses can regularize LLM/VLM planners to maintain and update a hypothesized dependency structure. Care must be taken to avoid revealing instance-specific solutions (e.g., by training only on procedurally generated sets disjoint from held-out seeds and by reporting generalization to unseen seeds).

**Limitations.** Point-and-Click currently focuses on single-room, 2D, click-only interactions with deterministic dynamics. While this isolates reasoning and perception, it omits physics-heavy manipulation, continuous control, and multi-room navigation. Visuals are synthetic and may privilege certain aesthetics; some clue types may induce bias if not balanced. Optimality depends on a reference policy that, although computed from the graph, might not be unique; this can penalize benign detours. Finally, agents could overfit to UI layout or renderer regularities; to mitigate this, we vary seeds, layouts, and assets, and we release a fixed, held-out evaluation suite alongside protocols for generating unlimited fresh instances.

## 6 CONCLUSION

Point-and-Click is a procedural benchmark of controllable DAG-structured puzzles (Key–Lock, Code–Lock, Pattern Match) with per-instance causal graphs for fine-grained evaluation. Results exposes a large human–agent gap, where models achieve partial progress without reliable completion, indicating a need for tighter perception–reasoning integration, persistent memory, and structure-aware planning. With scalable diversity, solvability, and analyzable structure, Point-and-Click provides a compact yet rigorous testbed for agents that perceive, remember, and reason.

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

# A APPENDIX

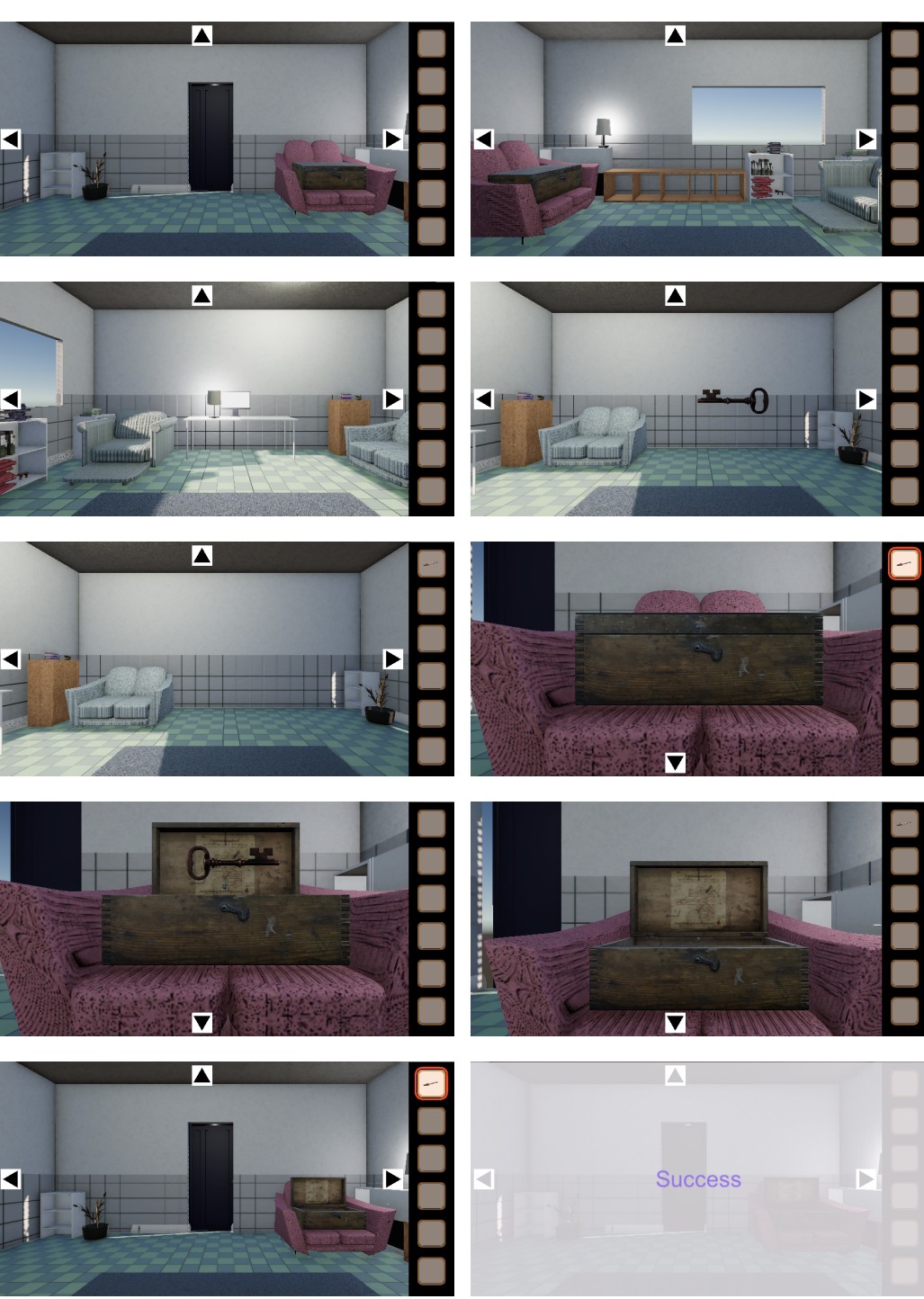

Figure 4: Annotated example for Claude-Sonnet-4.5 solving key-lock puzzle in Section 4.3 Case 1.

