# OpenReview forum: "Point-and-Click: A Procedural Benchmark for 2D Adventure Puzzle Solving"
_ICLR.cc/2026/Conference — Submitted to ICLR 2026_

### Official Review · Reviewer_6EdQ · 2025-10-31

**Soundness:** 2
**Presentation:** 3
**Contribution:** 2
**Rating:** 4
**Confidence:** 2

**Summary:**

The paper proposes Point-and-Click, a procedurally generated benchmark for evaluating multimodal agents on long-horizon, goal-inference puzzle solving in 2D adventure-game environments. Each puzzle is represented as a DAG that defines the dependencies between sub-tasks. The benchmark currently supports click-only interactions. By evaluating 3 different agents, the results reveal a significant performance gap between agents and humans, especially on hard set, where all 3 agents achieve a 0% success rate.

**Strengths:**

- The framework leverages DAGs to generate puzzles, allowing difficulty control by adjusting the number of steps, DAG size, and branching factors.
- Unlike standard datasets with clearly defined goals or ground-truth answers, this benchmark does not have an explicit goal. Instead, it requires agents to explore the environment and recognize what constitutes success, which makes it stand out from standard datasets.

**Weaknesses:**

-  Although Section 4 describes error types and includes a subsection on case studies, no actual visual examples are provided in the main paper, only textual summaries. The text references the appendix, but the appendix and supplementary material are not available in the submission. Including a few annotated examples would make the diagnostics clearer.
- The paper spends most of its space describing the framework and implementation details, which is good, but some of this content could be moved to the appendix to leave more space for the experimental setup and results analysis. I found the current experimental setup a little vague, and the experimental results do not provide many insights.
- The framework relies on a limited set of puzzle primitives (3 main primitives). While this makes puzzle generation controllable and analysis easier, it also restricts the logical diversity of the benchmark. Even with longer DAGs, the underlying reasoning patterns might remain relatively simple and repetitive.
- A small suggestion, Figure 1(b) font is too small.

**Questions:**

- Could the authors elaborate on the human baseline setup? For example, how many participants were involved, what was their background, and what was their experimental setting? Were the human participants also restricted to a given action budget?
- The paper mentions 30 puzzles across 3 difficulty levels, each ranging from 10 to 100 steps. How is the difficulty determined in this case? Is it defined strictly by the average step count, or also by other factors such as graph branching and dead-end density?

---

> ### Author Response · Authors · 2025-12-03
>
> (1) We thank the reviewer for the valuable advice. We have now added one annotated examples for case study in the appendix, which helps better visualize the trajectory and demonstrate the diagnostic, and will add more in the camera-ready version.
>
> (2) We thank the reviewer for the advice. We have now restructured the paper, removing part of descriptions of the benchmark and adding more details about the experiment setup. For more experimental insights, please see answer (2) to Reviewer rC76 and answer to Reviewer LDGJ.
>
> (3) We would like to clarify that this three puzzle primitives are **abstraction** for all kinds of puzzles in rich commercial point-and-click games, and does not imply that the resulting reasoning patterns are simple or repetitive. Take the famous 2D point-and-click series Rusty Lake as an example. Most of its puzzles can be categorized into the same three interaction types as in our environment:
>
> - Require an object (inventory-dependent): puzzles including finding a key, multiple pieces for a complete photo, alchemy with a given recipe, etc.. These correspond to our Key–Lock primitive, where “key” is an abstract notion for all kinds of puzzle elements.
> - Require a visual clue but no object (memory-heavy): puzzles including clock position, rotation dials, or sequence on one TV screen that must be remembered and later entered elsewhere. This matches our Code–Lock primitives where the challenge is cross-view perception, working memory, and mapping visual hints to the correct symbolic action.
> - Require neither external clue nor object (self-contained logic): Rusty Lake also includes local puzzles such as sliding tiles, frog jumps, or within-view pattern rearrangements that can be solved purely by local reasoning. These are captured by our self-contained Pattern Match puzzles, which do not depend on inventory or long-range clues.
>
> Despite being built from these few categories, Rusty Lake games are widely regarded as having rich, varied, and quite challenging logic. The diversity arises from how these step types are expressed with specific puzzle elements and how they are chained and interleaved. Empirically, current state-of-the-art agents already struggle significantly on this space, indicating that the present primitive set is far from saturating their capabilities.
>
> (4) Thanks for the suggestion. We have enlarged the fonts. It should be clearer now.
>
> (5) Human baselinse setup: there are 10 participants, all undergraduate / graduate students. Their settings are exactly the same as the models, with visual input and click interaction only. They receive a same budget of 20 times of ground-truth steps for the puzzle as models do, but they never use up the budgets.
>
> (6) These 30 puzzles have gone through manual inspection. Therefore the difficulty is determined by both step counts and other factors such as graph branching, starting position, etc.

---

### Official Review · Reviewer_CQ8e · 2025-11-01

**Soundness:** 3
**Presentation:** 3
**Contribution:** 3
**Rating:** 8
**Confidence:** 4

**Summary:**

The paper introduces Point-and-Click, a benchmark for evaluating the ability of vision-language models to perform well on point-and-click puzzle games, testing long-horizon planning, grounded reasoning, and extrapolating world knowledge. The authors create a system for procedurally generating huge amounts of puzzles, spanning up to hundreds of steps with complex compositional dependencies (locks, keys, combinations, etc), and having no explicitly stated goal; e.g., the model must itself figure out what actions to take in order to complete the puzzle. The authors evaluate the performance of frontier models, discuss the types of errors that often occur, and explore the implications of the results on model design.

**Strengths:**

This is a great benchmark!

1. It measures an important ability of models in an interesting and intuitive way;
2. Procedural generation guarantees that there's no dataset contamination, a major issue with similar benchmarks;
3. The ground-truth causal graph allows evaluation of partial completion, which isn't the case for many other puzzle-related benchmarks.

The paper is clearly written, well-motivated, and describes the bechmark well. The evaluations are well-done and highlight the usefulness of point (3) above.

**Weaknesses:**

I think this paper lacks significant weaknesses; of course, it would always be nice if the benchmark was more diverse (e.g. more complex puzzle primitives, more 3D reasoning, etc), but this is trivially a weakness of every benchmark; the dataset introduced in the paper is good and logically self-contained.

Minor comments

- On lines 103–104, the performance is broken down into three categories, but I don't think the categories have been defined yet.

**Questions:**

As this is a benchmark paper that introduces its benchmark very clearly, I don't have any immediate questions for the authors.

---

> ### Author Response · Authors · 2025-12-03
>
> We appreciate the reviewer for the positive and encouraging review. We are glad that you find the benchmark setup intuitive, the procedural generation helpful for avoiding data contamination, and the use of the ground-truth causal graph valuable for evaluating partial completion.
>
> Regarding benchmark diversity, it is indeed trivially a weakness of every benchmark. Since Point-and-Click is targeted for active maintenance, and the benchmark structure is modular, we may incorporate richer puzzle primitives in the future.
>
> For the minor comment: thanks for the reminder! The three categories are Simple, Medium, Hard, as reported in Table 1 in Section 4.2 Benchmarking Results. We have now added a sentence explaining this.

---

### Official Review · Reviewer_LDGJ · 2025-11-01

**Soundness:** 2
**Presentation:** 3
**Contribution:** 3
**Rating:** 4
**Confidence:** 4

**Summary:**

Point-and-Click introduces a procedurally generated benchmark for multimodal agents to solve 2D point-and-click adventure puzzles with implicit goals, long-horizon dependencies, and language-perception grounding. Each game instance is built from controllable DAG-structured puzzle graphs, rendered as interactive rooms, and comes with ground-truth causal graphs for fine-grained evaluation. Experiments show a large human–agent gap: state-of-the-art computer-use/VLM agents achieve low success while humans remain high, revealing failures in perception/attention, riddle solving, memory, and long-horizon planning.

**Strengths:**

1. Tests GUI agents on implicit-goal puzzle solving (agent must infer what “success” even is), which is distinct from instruction-following paradigms.
2. Reports success, step completion, optimality vs. reference plan, and error categories (perception/attention misses, riddle-reasoning, forgetting clues) are useful diagnostics beyond accuracy.

**Weaknesses:**

1. The main tables lack uncertainty bars and budget-sensitivity (timeouts vs. success); it’s unclear how robust rankings are to interaction budgets and prompt seeds.
2. The paper evaluates only a small set of agents (two commercial “computer-use” systems and one open-source GUI agent), which makes it hard to separate paradigm effects from vendor-stack biases, and limits confidence that conclusions generalize across model families. I highly recommend the authors to add results in Qwen2.5-VL-72B-Instruct, InternVL 2.5/3, Llama-3.2-Vision ....

**Questions:**

I recommend bucketing instances by puzzle primitives:Key–Lock, Code–Lock, Pattern-Match, and their compositions, and reporting per-bucket Success and Efficiency@Success (optionally add last-3-step failure rate, input/typo errors for Code–Lock, and localization misses for Pattern-Match). This single table will localize whether performance drops are driven primarily by reasoning/planning & memory (Key–Lock), symbolic extraction and precise entry (Code–Lock), or visual perception/targeting (Pattern-Match), thereby pinpointing which capability family is the bottleneck and guiding targeted model/ablation improvements.

---

> ### Author Response · Authors · 2025-12-03
>
> We greatly appreciate the reviewer's suggestion to (1) include uncertainty bars and budget-sensitivity for more robust evaluation (2) include more model families (3) bucketing instances by puzzle primitives.  To incorporate VLMs into our environment, we ask the models to directly output interaction coordinates as well as descriptions of the scene in json format. We replace OpenAI / Claude models with GPT-5.1 and Claude-Sonnet-4.5 using this same format. The new experiments are as follows:
>
> (1) Key-Lock
>
> | Models   | Success@50 | Success@100 | Success@200 | Subgoals    | Efficiency  |
> | -------- | ---------- | ----------- | ----------- | ----------- | ----------- |
> | GPT      | 0% (±0.0)  | 0% (±0.0)   | 0%(±0.0)    | 1/6 (±0.0)  | N/A         |
> | Claude   | 0% (±0.0)  | 50%(±0.25)  | 100% (±0.0) | 6/6  (±0.0) | 9.9 (±4.53) |
> | Qwen     | 0% (±0.0)  | 0% (±0.0)   | 0% (±0.0)   | 0/6 (±0.0)  | N/A         |
> | InternVL | 0% (±0.0)  | 0% (±0.0)   | 0% (±0.0)   | 0/6 (±0.0)  | N/A         |
> | Llama    | 0% (±0.0)  | 0% (±0.0)   | 0% (±0.0)   | 0/6 (±0.0)  | N/A         |
>
> (2) Code-Lock
>
> | Models   | Success@50 | Success@100 | Success@200 | Subgoals   | Efficiency |
> | -------- | ---------- | ----------- | ----------- | ---------- | ---------- |
> | GPT      | 0% (±0.0)  | 0% (±0.0)   | 0 %(±0.0)   | 2/6 (±0.0) | N/A        |
> | Claude   | 0% (±0.0)  | 0% (±0.0)   | 0 %(±0.0)   | 5/6 (±0.0) | N/A        |
> | Qwen     | 0% (±0.0)  | 0% (±0.0)   | 0 %(±0.0)   | 0/6 (±0.0) | N/A        |
> | InternVL | 0% (±0.0)  | 0% (±0.0)   | 0 %(±0.0)   | 1/6 (±0.0) | N/A        |
> | Llama    | 0% (±0.0)  | 0% (±0.0)   | 0 %(±0.0)   | 1/6 (±0.0) | N/A        |
>
> (3) Pattern Match
>
> | Models   | Success@50 | Success@100 | Success@200 | Subgoals     | Efficiency |
> | -------- | ---------- | ----------- | ----------- | ------------ | ---------- |
> | GPT      | 0% (±0.0)  | 0% (±0.0)   | 0% (±0.0)   | 0 / 3 (±0.0) | N/A        |
> | Claude   | 0% (±0.0)  | 0% (±0.0)   | 0% (±0.0)   | 0 / 3 (±0.0) | N/A        |
> | Qwen     | 0% (±0.0)  | 0% (±0.0)   | 0% (±0.0)   | 0 / 3 (±0.0) | N/A        |
> | InternVL | 0% (±0.0)  | 0% (±0.0)   | 0% (±0.0)   | 0 / 3 (±0.0) | N/A        |
> | Llama    | 0% (±0.0)  | 0% (±0.0)   | 0% (±0.0)   | 0 / 3 (±0.0) | N/A        |
>
> (4) Combination
>
> | Models   | Success@50 | Success@100 | Success@200 | Subgoals    | Efficiency |
> | -------- | ---------- | ----------- | ----------- | ----------- | ---------- |
> | GPT      | 0% (±0.0)  | 0% (±0.0)   | 0% (±0.0)   | 0 /  (±0.0) | N/A        |
> | Claude   | 0% (±0.0)  | 0% (±0.0)   | 0% (±0.0)   | 0 / (±0.0)  | N/A        |
> | Qwen     | 0% (±0.0)  | 0% (±0.0)   | 0% (±0.0)   | 0 /  (±0.0) | N/A        |
> | InternVL | 0% (±0.0)  | 0% (±0.0)   | 0% (±0.0)   | 0 /  (±0.0) | N/A        |
> | Llama    | 0% (±0.0)  | 0% (±0.0)   | 0% (±0.0)   | 0 /  (±0.0) | N/A        |
>
> These experiments are listed in ascending order of difficulty. The reviewer's suggestion, combined with reasoning trace in response (2) to Reviewer rC76, greatly helped us to localize the main driving factor for performance drop: visual perception/targeting (Pattern-Match). All the models completely fail at this puzzle primitive. On the higher level, they fail to infer the underlying goal of pattern-match. Most of them infer it's about finding a missing piece to make the picture complete, instead of interpreting it as sliding puzzles. On the lower level, even if with explicit hint that this is a sliding puzzle, they still cannot reason a correct action trace, even if the ground-truth solution is only 3 steps. This observation indicates a fundamental drawback in visual reasoning in current VLMs.
>
> We also observe distinct behaviors from different models. Claude-Sonnet-4.5 generally has the best performance under the new setting, which successfully perceive the scene view, reason about puzzle solutions, and output precisely grounded coordinates to operate with the game interface. GPT-5.1 is good at exploring all the interactive objects, but struggles with precise grounding (e.g. correctly clicking on the buttons to nevigate around the room). Open-source models such as Qwen2.5-VL-72B-Instruct, InternVL 2.5/3, Llama-3.2-Vision even fail to successfully navigate around the room or inspect any interactable objects, let alone reasoning about the solution and making meaningful progress.
>
> All these experimental observations show that Point-and-Click poses significant challenges to current VLMs. They validate the value of Point-and-Click in effectively revealing the limitations of current VLMs in implicit goal inference, compositional reasoning, and language-perception grounding, which we believe is a valuable contribution to the existing community.

---

### Official Review · Reviewer_rC76 · 2025-11-02

**Soundness:** 3
**Presentation:** 3
**Contribution:** 3
**Rating:** 6
**Confidence:** 4

**Summary:**

The paper presents a procedurally generated 2D point-and-click puzzle benchmark to evaluate multimodal agents on long-horizon, visually grounded planning with implicit goals. The task seem to be far from existing paradigm of agent evaluations. Tasks are posed as dependency-DAG escape puzzles, solved through pixel-click interaction. The system provides ground-truth graphs, controllable difficulty, and fine-grained metrics. Experiments show large human–agent gaps.

**Strengths:**

- the overall design of the task is fairly innnovative, existing paradigm does not seem to exist.

- The procedural generation with DAG can avoid data contamination for future evaluations, it also offers the opportunity of structured metrics.

- the study also introduces human evaluations to demonstrate the gap.

**Weaknesses:**

- There are only 30 puzzles tested (10 per tier). The number seem too few to claim generalization across a procedural domain.

- The benchmark mixes visual recognition and planning without controlled ablations, although there are some analysis to infer where the procedure fails, some additional versions of the benchmark that evaluates part of the process can be valuable.

- It is not so clear whether the puzzles will allow alternative solutions (even generated with a underlying DAG), is there a possiblity that a puzzle can be solve by other paths different from the underlying DAG?

- The paper does not seem to use this oppoturnity to clearly discuss the strengths and limitations under this new evaluations.

**Questions:**

please address the limitations discussed.

---

> ### Author Response · Authors · 2025-12-03
>
> We thank the reviewer for recognizing the novelty of Point-and-Click, its procedural generation scheme, and inclusion of human study.
>
> (1) Regarding the number of 30 puzzles (10 per tier), we were following the scale of previous work, such as EscapeCraft (11 scenes for each of Difficulty-1 and Difficulty-2, and 21 scenes for Difficulty-3), VisEscape (20 escape rooms in total), and FlashAdventure (34 adventure games in total, with 15 puzzles for room escape). If the reviewer deems it necessary to increase scale to 100 per tier, we are happy to include those extensive experiments in camera-ready version.
>
> (2) We initially introduce this benchmark structure and experiment setting because we found it hard to distinctly distinguish between visual recognition and planning, as visual perception is essential for recognizing the affordance of objects, therefore crucial to reasoning about the puzzle. As an attempt to separate the visual recognition with planning, we now explicitly require the model to output their observations of the scene, and their rationale for choosing the coordinates for interaction. Manual inspections show that models are pretty good at recognizing game objects. Although the level of detail varies, the models can generally correctly identify game-related important objects and partially, appropriately reason about the game logic. Experimental observations reveal that a major difficulty lies in **grounding**, namely how to correctly interact with the game interface according to a planned strategy, including picking up keys, clicking buttons to switch room views, and so on. The models often have the right high-level idea, but exhibit substantial shortcomings in accurately clicking on objects.
>
> (3) Given the underlying DAG, the solution is unique in the sense that all previous nodes (incoming edges) must be satisfied before proceeding to the next node, but the order of solving these previous subgoals can vary.
>
> (4) We tried to discuss the strengths and limitations under this new evaluations in Section 4.3 Case Study and Section 5 Discussion. Additional study in (2) reveals the strength in visual recognition, basic level planning, but limitation in grounding and reasoning about complicated puzzles. For more experimental insights, please also refer to answer to Reviewer LDGJ.

---

### Meta-Review · Area_Chair_gHES · 2026-01-04

**Summary:**

The paper presents a procedural benchmark for 2D adventure games that generates puzzles via controllable directed acyclic graphs (DAGs). This approach is highly effective at preventing data contamination, a major issue for existing benchmarks. However, the current iteration of the paper is hampered by a very small evaluation scale and a narrow range of puzzle logic that may not yet provide a robust measure of general-purpose reasoning.

Another concern is that while the benchmark is not explicitly designed to force failure across all models, the authors’ analysis of failure cases lacks the necessary depth to be fully comprehensive. As promised during the rebuttal, the authors should substantially increase the number of puzzle instances to ensure statistical significance. Furthermore, I believe the authors should conduct a more granular investigation into "Easy" level tasks, providing a clearer classification within this tier to better illustrate the specific functional boundaries of current models. Finally, the evaluation remains incomplete without the inclusion of Gemini models; comparing these state-of-the-art agents alongside the existing commercial and open-source baselines is essential for a truly representative assessment of the current landscape. The AC encourages the author to further improve their paper and submit to the next venues.

**Reviewer Concerns:**

Following Reviewer LDGJ’s suggestion, they bucketed results by primitive type, which provided the valuable insight that current VLMs struggle most with Pattern-Match puzzles due to a lack of fundamental perception ability.

Reviewer 6EdQ's concerns regarding the human comparison were addressed by detailing a baseline of 10 participants who operated under the same constraints as the AI agents.

Reviewer rC76 correctly identified that testing only 30 puzzles is insufficient for a procedural benchmark meant to demonstrate generalization. While the authors offered to scale to 100 per tier, this data was not integrated into the current version, leaving the benchmark’s statistical power in question.

Reviewer 6EdQ argued that with only three main primitives, the reasoning patterns remain repetitive. The authors’ comparison to the Rusty Lake game series was noted, but the benchmark’s current logical depth remains quite narrow.

To Rebuttal experiments showed that models often fail at low-level "grounding" (clicking the right UI coordinates) rather than high-level reasoning. This suggests the benchmark may currently be measuring GUI control limitations as much as cognitive puzzle-solving.

**Reviewer Scores:**

Reviewer rC76: 6, likely lower or maintain the score, as the paper does not provide enough details and analysis to give more insights from this new benchmark.
Reviewer LDGJ: 4, the additional experiments provided by the authors do not include Gemini 2.5, and the results on the VLM models somehow do not fit in this benchmark.
Reviewer CQ8e : 8
Reviewer 6EdQ : 4

---

### Decision · Program_Chairs · 2026-01-26

Reject